# 3200 ppi Matrix-Addressable Blue MicroLED Display

**DOI:** 10.3390/mi13081350

**Published:** 2022-08-19

**Authors:** Meng-Chyi Wu, Ming-Che Chung, Cheng-Yeu Wu

**Affiliations:** 1Institute of Electronics Engineering, National Tsing Hua University, Hsinchu 30013, Taiwan; 2Department of Surgery, Division of Plastic and Reconstructive Surgery, Veterans General Hospital, Taichung 40705, Taiwan

**Keywords:** active matrix, MicroLED, micro-display, high pixels per inch, flip-chip bonding

## Abstract

In this article, an active matrix (AM) micro light-emitting diode (MicroLED) display with a resolution of 1920 × 1080 and a high pixel density of 3200 pixels per inch (ppi) is reported. The single pixel with a diameter of 5 μm on the MicroLED array exhibits excellent characteristics, including a forward voltage of 2.8 V at 4.4 μA, an ideality factor of 1.7 in the forward bias of 2–3 V, an extremely low leakage current of 131 fA at −10 V, an external quantum efficiency of 6.5%, and a wall-plug efficiency of 6.6% at 10.2 A/cm^2^, a light output power of 28.3 μW and brightness of 1.6 × 10^5^ cd/m^2^ (nits) at 1 mA. The observed blue shift in the electroluminent peak wavelength is only 6.6 nm from 441.2 nm to 434.6 nm with increasing the current from 5 μA to 1 mA (from 10 to 5 × 10^3^ A/cm^2^). Through flip-chip bonding technology, the 1920 × 1080 bottom-emitting MicroLED display through the backside of a sapphire substrate can demonstrate high-resolution graphic images.

## 1. Introduction

The emerging micro light-emitting diodes (MicroLEDs) are anticipated for the next-generation displays, especially augmented reality (AR) or virtual reality (VR) as well as for wearable, portable devices [1,2,3,4,5,6,7,8,9,10,11], visible light communication (VLC) [12,13,14,15,16], and optogenetics [17,18]. The self-emissive MicroLEDs can generate bright images efficiently without external light sources and lossy optical components. In addition, the MicroLEDs are more advantageous in terms of brightness, lifetime, thermal stability, and robustness in extreme conditions. For display applications, image quality on display is determined to a large extent by pixel density. The smaller the pixels and higher pixel density, the less likely the viewer can perceive pixelation, whereby images become photorealistic, and fonts can be formed more perfectly as pixel size decreases [19]. For the application of near-eye displays or head-mounted wearable displays, specifications required for ideal penetrating smart glasses: (1) the size of the display light engine should be controlled below about 1 inch; (2) due to the requirements of content recognition, the brightness must be above 4000 nits to ensure that it is not affected by external environments such as weather or venue; (3) the resolution should also be above 3000 ppi (pixels per inch) so that the projected enlarged picture is clear enough.

For the decade development, the active matrix (AM) MicroLED displays have been widely researched [20,21,22,23,24,25,26,27,28,29] because the AM driving schemes are more preferable for micro-displays with high resolution as compared to the passive matrix (PM) schemes [30,31]. AM provides an independent driver for each pixel, greatly enhancing the driving capacity. Jiang et al. [20] were the first to research the possibility of using MicroLEDs to produce AM with a 10 × 10 resolution at a pixel pitch of 50 μm. Zhang et al. [23] fabricated 635 ppi blue AM micro-displays with 64 × 36 resolution at a pitch of 40 μm. Furthermore, Day et al. [24] successfully presented the first high-resolution blue and green AM micro-displays with a resolution of 640 × 480 or 1700 ppi over a pixel diameter of 12 μm and a pixel pitch of 15 μm. Dupré et al. [27] developed a blue and green AM display with resolutions of 873 × 500 or 2500 ppi across a diameter of only 8 μm at a pixel pitch of 10 μm. Moreover, our previous work [28] also reached a 960 × 540 micro-display with a resolution of 2000 ppi. In addition to these GaN-on-sapphire micro-displays mentioned above, the blue GaN-on-Si micro-display consists of 400 × 240 pixels with a pitch size of 20 μm × 20 μm and 848 ppi within a display diagonal of 0.55 inches and was reported by Qi et al. [21]. Liang et al. demonstrate a 0.55-inch full-color micro-LED display with 992 ppi resolution. Their display was structured with a blue micro-LED array of 1984 ppi which was bonded on a CMOS driving matrix, and it was converted from a monochrome display to a full-colored one by colloidal quantum dots [32].

For the MicroLED fabrication, the pixel is always etched in the mesa shape with a vertical sidewall. However, as the pixel size scales down, it is very difficult to control the wet etching of ITO film on the top of the device structure for the vertical sidewall, and the grid-electrode lithographic process becomes a challenge. In order to fabricate a high-pixel-density microdisplay, novel self-aligned technology is developed to etch the pixel as an inverted trapezoidal-shape mesa by adjusting the dry etching conditions. In this work, we present a 1920 × 1080 AM micro-display with a frame of 16:9 aspect ratio. A single pixel size is 5 μm, and a pixel pitch of 8 μm within a diagonal length is 0.69 inches, and hence, the quite high 3200 ppi is obtained. The fabricated micro-display is bottom emissive through the sapphire substrate and hybridized on a CMOS active matrix by using high precision flip chip bonder. Each pixel of the matrix exhibits a remarkably low leakage current of 131 fA at −10 V, a low forward voltage (V_F_) of 2.8 V at the current density of 22.2 A/cm^2^, and high external quantum efficiency (EQE) of 6.5% at the current density of 10.2 A/cm^2^.

## 2. Experimental Section

In order to fabricate the high-resolution micro-display and such a precise pattern, the device process was used on the commercial 2-inch InGaN/GaN blue LED wafer grown on sapphire substrate in this work, as shown in step 1 of Figure 1. The novel self-aligned mesa formation is schematically described in Figure 1. First, the pixels of the array were defined via a lithographic aligner SUSS-MJB4. Then, a 70 nm indium tin oxide (ITO) was deposited by RF sputter onto the wafer as the ohmic contact layer of p-type GaN, and an 80 nm Ni layer was deposited by e-beam evaporation as the dry-etching hard mask, as shown in step 2 of Figure 1. The pixels were etched by inductively coupled plasma-reactive ion etch (ICP-RIE) afterward. In this step, the shape of cross-sectional profile of the mesas was in upside-down trapezoid after ICP-RIE etching. The etched parameters consisted of a bias power of 500 W, a Cl_2_ flow rate of 180 sccm, an ICP power of 20 W, a pressure of 2.5 Pa, an ambient temperature of 60 °C, and an etching time of 4 min. The estimated etching rate is about 7.5 nm/sec. The images by scanning electron microscopy (SEM) of the upside-down trapezoid etching profile of the pixels are shown in Figure 2, whose sidewall tilts at around 80 degrees angle. The structure makes the metals not connect between the top of the pixels and the bottom n-GaN region, efficiently screening the sidewall from the deposited metals, as shown in step 3 of Figure 1. Because of the self-aligned technology, the common cathode and pixel anode of 30/120/40/60 nm Ti/Al/Ti/Au were directly deposited onto the chip without any lithographic process onto the n-GaN region and pixel region, respectively, as shown in step 4 of Figure 1. The self-aligned process allows the fabrication to complete efficiently on such small size device, greatly improving the alignment precision and uniformity. The 100 nm Al_2_O_3_ dielectric layer was then passivated by plasma-enhanced atomic layer deposition (PE-ALD) to reduce leakage current from the dry-etching damaged sidewall of the pixels. Indium bumps were deposited on the top of each metal electrode. The single indium bump has an average resistance of about 67 mΩ tested from the designed daisy chain [29]. Finally, the micro-LED array was hybridized on a CMOS active matrix model JD2702 from Jasper Display Corporation by using high precision flip chip bonder. Illustration of SEM photograph of the cross-section with indium bumps integrated with the active matrix CMOS platform for the 1920 × 1080 micro-display is presented in Figure 3.

## 3. Results and Discussion

Figure 4 shows the current as a function of voltage (I−V) for a single pixel of the blue 1920 × 1080 micro-display. The V_F_ is determined by the relationship between the injected current and the LED active area, which is usually defined at the driven current of 20 mA with an active area of 300 μm × 300 μm, corresponding to a current density of 22.2 A/cm^2^ (or 4.4 μA). Thus, the V_F_ of a single pixel is about 2.8 V at 4.4 μA. This measured current ranging from 5 μA to 1 mA is very small for the MicroLED pixel. Therefore, the heatsink is not required. In addition, the series resistance above 4 V and ideality factor in the forward bias of 2–3 V of a single pixel of the 1920 × 1080 micro-LEDs was calculated as 901 Ω and 1.7, respectively. This higher series resistance is attributed to the smaller MicroLED size and the designed n electrode as a grid-like pattern [29], as shown in Figure 2b. Another reason for the high series resistance is mainly due to the possibility that the inverted trapezoid structure must be evaporated with ITO/Ni as a hard mask before etching, and the bonding to the LED structure must be enhanced by the RTA process before dry etching. The subsequent deposited p/n metals do not undergo the second high-temperature RTA process to avoid the scorch of Ni metal. Thus it leads to higher series resistance. The reverse current of a single pixel at −5 V has an astonishingly small value of 131 fA or 6.7 × 10^−^^7^ A/cm^2^, presenting excellent electrical characteristics for a single pixel. This extremely low reverse leakage current is much lower than those of 5 pA at −5 V for the MicroLEDs with a size of 20 × 20 µm^2^ [21,23], 1 pA at −3 V for the MicroLEDs with a size of 20 × 20 µm^2^ [33], and 0.1 nA for the MicroLED with a size of 25 × 25 µm^2^ by treating the sidewall with PECVD-SiO_2_ [34].

Figure 5 illustrates the plot of EQE and wall-plug efficiency (WPE) as a function of logarithmic current density for a single pixel size of 5 μm for the 1920 × 1080 blue MicroLEDs. The inset shows the linear characteristics of EQE and WPE versus injected current for a single pixel. It is worth noting that the EQE was directly measured on the MicroLED chip through the probe station; it is possible to obtain a lower EQE value compared to EQE measured within an integrating sphere. The estimated EQE reaches 5.0% at 100 A/cm^2^ and the highest at 6.5% at 10.2 A/cm^2^. The estimated maximum WPE occurs at about 6.6% at 10.2 A/cm^2^. Based on the size-reduction in the performances of GaN-based MicroLEDs, Olivier et al. [35] and Daami et al. [36] also measured the EQE from the sapphire side. The peak value of EQE reached 5%, reported in Ref. [35] for 10-μm-pixel MicroLED. Meanwhile, its peak EQE also appeared at the current density of around 30 A/cm^2^. Moreover, Templier [37] also reported the 6.5 μm × 6.5 μm blue MicroLED has a maximum EQE of 9.5%, obtained at a current density of 20 A/cm^2^ (or 25 μA). Tian et al. reported a similar EQE value for the 6 μm pixel of blue MicroLED [38], but the peak current density occurred at 160 A/cm^2^, which is much higher than our 10.2 A/cm^2^. Figure 6 shows the light output power, luminous flux, and brightness as a function of injected current for a single pixel of 1920 × 1080 MicroLEDs. All the light output power, luminous flux, and brightness almost linearly increase with injection current and reach 28.3 µW, 0.4 mlm, and 1.6 × 10^5^ cd/m^2^ (nits) at 1 mA, respectively. The 5 × 5 μm^2^-sized MicroLED pixel has a maximum light output power of 1.4 W/mm^2^ at 1 mA. The outstanding brightness level is several orders of magnitude higher than those of LCDs and OLEDs. Figure 7 shows the peak wavelength and full width at half maximum (FWHM) of electroluminescence (EL) spectra as a function of injection current density for a single pixel. It shows a peak wavelength at around 441.2 nm and a full width at half maximum (FWHM) of 13.1 nm at 5 μA. It is observed that the blue shift in the EL peak wavelength is only 6.6 nm from 441.2 to 434.6 nm with increasing the current from 5 μA to 1 mA resulting from the shielding of the polarized electric field in associated quantum confined stark effect (QCSE). The FWHM increases from 13.1 nm at 5 µA to 29.4 nm at 1 mA. The FWHM broadening results from the fluctuations of indium content and thus the bandgap in the InGaN quantum wells of the active region, which cause the different potential minima [39]. Figure 8 presents the original image sources of a high-resolution leopard and its corresponding images, respectively. The CMOS driver IC can only supply 250 mA to the MicroLED display at near the forward voltage of 2.7 V. The 250 mA would support the current to 1920 × 1080 (=2,073,600) pixels, and every single pixel shares the current of 0.12 μA on average. Therefore, no heat sink is considered in the microdisplay. We demonstrate the AM blue 1920 × 1080 micro-displays with a high resolution of 3200 ppi and an aspect ratio of 16:9, a small pixel pitch of 8 μm, and a small pixel of 5 μm.

## 4. Conclusions

We have demonstrated a 1920 × 1080 blue AM MicroLED display using a novel self-aligned method. Such small pixel and pixel-pitch can be reached by using the novel self-aligned technology, showing the promising realization for the fabrication of high resolution, high ppi, and self-emissive displays. The AM high-resolution 1920 × 1080 blue micro-display consists of 5 µm pixel size, 8 µm pixel pitch, 3200 ppi, and 0.69-inch diagonal length. The MicroLED pixel exhibits excellent electrical and optical characteristics, including a low leakage current of 131 fA at −10 V, a low forward voltage of 2.8 V, a high EQE of 6.5%, and a WPE of 6.6% occurring at 10.2 A/cm^2^. Moreover, the light output power, luminous flux, and brightness reach 28.3 µW, 0.4 mlm, and 1.6 × 10^5^ cd/m^2^ (nits) at 1 mA, respectively. With these results, the MicroLED display is a prospective technology for high-resolution applications such as AR, VR, and compact projectors, such as these kinds of advanced high-resolution head-up display (HUD) systems/wearable displays.

## Figures and Tables

**Figure 1 micromachines-13-01350-f001:**
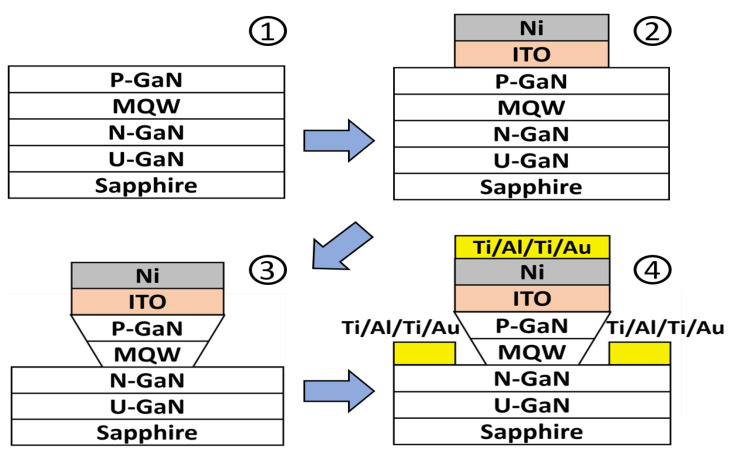
The schematic process steps of upside-down trapezoid mesa using the self-aligned process to avoid the metals depositing onto the mesa sidewall for the MicroLED array. At the last step, the common cathode and pixel anode can be directly deposited on the chip without lift-off process.

**Figure 2 micromachines-13-01350-f002:**
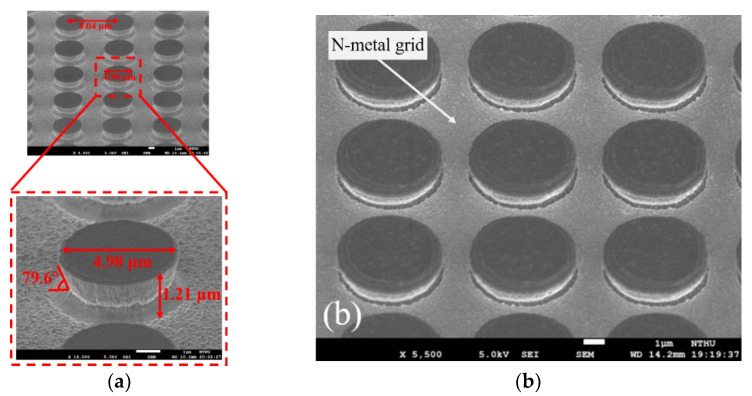
(**a**) The MicroLED array with the upside-down trapezoid mesa observed at 45° viewing angle via scanning electron microscope (SEM). The mesa diameter is 4.98 μm, the mesa height is 1.2 μm, and the sidewall tilts at an angle of 80 degrees. We can see the corner of the sidewall just like the projection of the mesa. (**b**) The SEM image of MicroLEDs with n-metal grid.

**Figure 3 micromachines-13-01350-f003:**
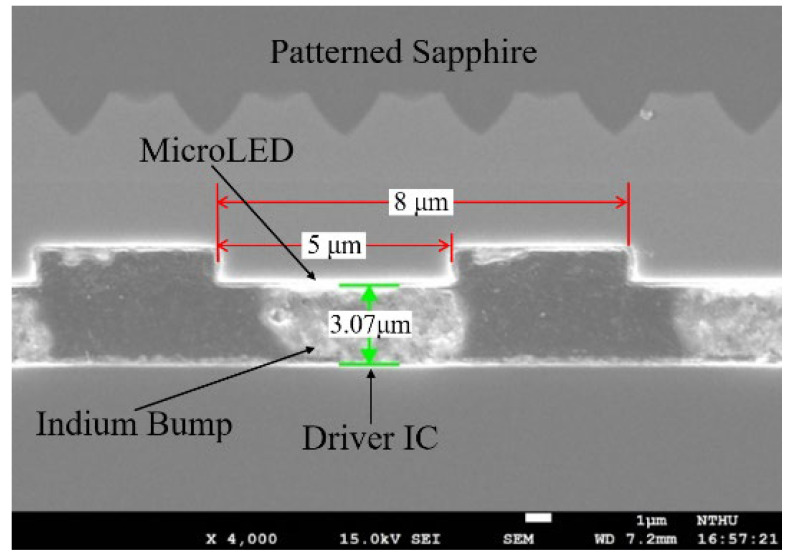
Cross section of flip-chip bonding between the indium bumps and the active matrix CMOS platform.

**Figure 4 micromachines-13-01350-f004:**
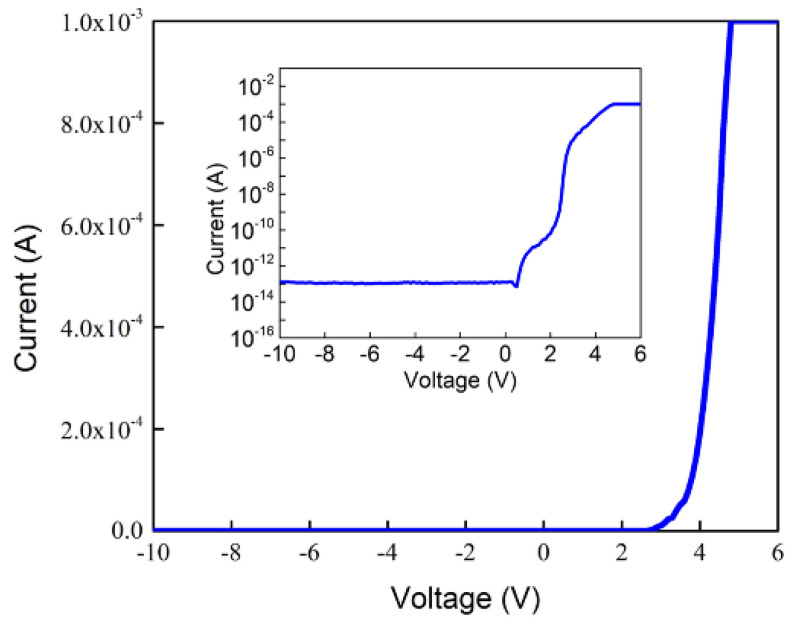
Linear current-voltage (I−V) characteristics for a single pixel with a 5 μm size for the 1920 × 1080 blue MicroLEDs. Inset: logarithm current versus voltage plot for a single pixel of blue MicroLEDs.

**Figure 5 micromachines-13-01350-f005:**
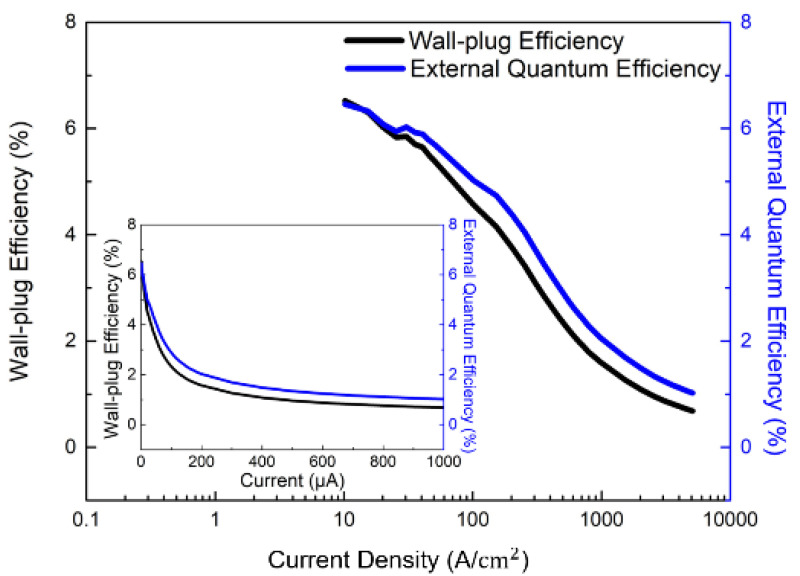
Plot of EQE and WPE as a function of current density in semi-log scale for a single pixel of the 1920 × 1080 blue InGaN MicroLEDs. Inset: Plot of EQE and WPE versus logarithmic current for a single pixel.

**Figure 6 micromachines-13-01350-f006:**
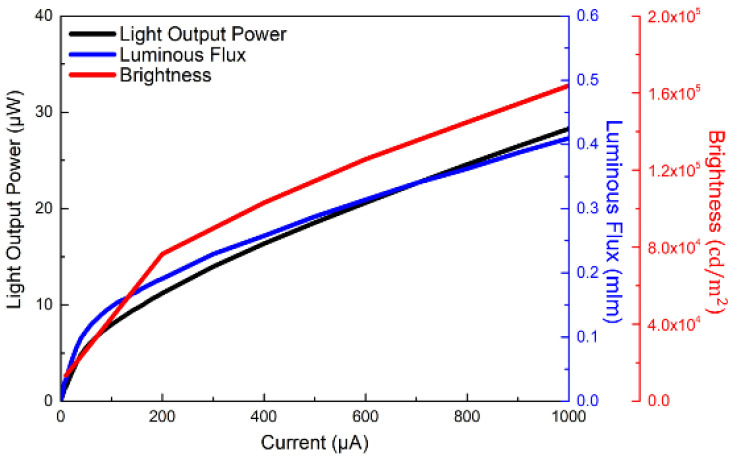
Plot of luminous flux, light output power, and brightness as a function of injection current for a single pixel of 1920 × 1080 blue MicroLED array.

**Figure 7 micromachines-13-01350-f007:**
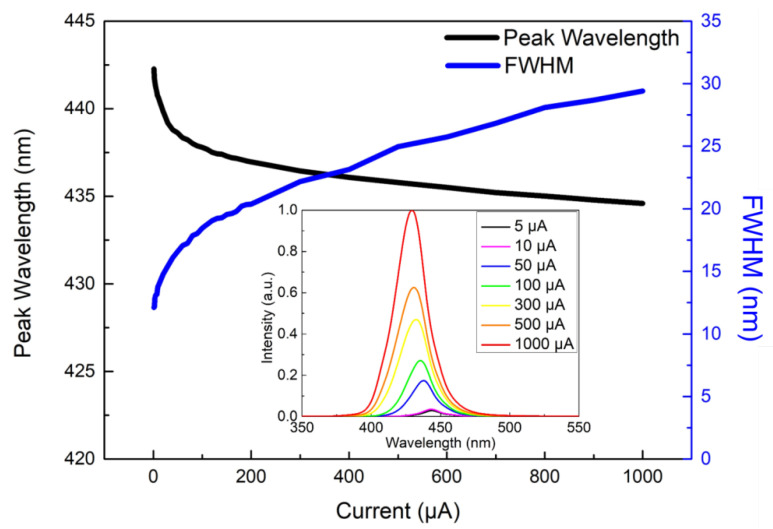
Peak wavelength and full width at half maximum (FWHM) as a function of injection current for a typical single pixel of the 1920 × 1080 blue MicroLED array. Inset: EL spectra at various forward currents from 5 μA to 1 mA.

**Figure 8 micromachines-13-01350-f008:**
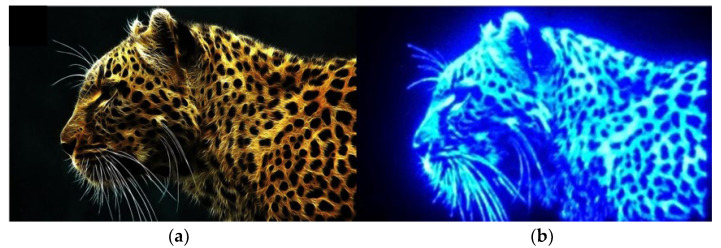
The original image (**a**) of a high resolution leopard and its corresponding image (**b**) on the 1920 × 1080 blue micro-display.

## Data Availability

Not applicable.

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
