# Peer review of "3200 ppi Matrix-Addressable Blue MicroLED Display"

_micromachines, 2022, doi:10.3390/mi13081350_

Round 1

Reviewer 1 Report

The authors have presented a very nice device and prototype level demonstration for high ppi blue monochromatic micro-LED display. The work is great but more information is needed for effective dissemination of this knowledge in the academic community-

1.       Line no. 68 – Please provide the detailed etch chemistry, wattage and material removal rate for the Ni, ITO and GaN etch to achieve the inverted trapezoidal etch profile and also ensure etch selectivity during each step.

2.       Line no. 79 – Please add another cross sectional schematic in figure 1 to show the dielectric layer conformality. Additionally showing the cross-section with indium bumps and integration with the active matrix CMOS platform will also be useful for the reader, otherwise it is difficult to appreciate the complexity of this process

3.       Line no. 106 - Can the grid-like patterned be properly explained with a figure? Any comments on the contact resistance? Have authors done a TLM study for the indium bumps?

4.       Line no. 123 – Could you please plot the EQE and WPE below 10 A/cm2?

5.       Line no. 136-138 - Could authors provide references for this statement?

6.       Line no. 142 - Please provide more details of your EQE measurement set-up, how much is the extraction efficiency for your devices, a few figures should be helpful to explain it to the reader. Typically absolute EQE is measured in an integrating sphere after the device is packaged with a resin to enhance the light extraction as much as possible. Therefore, for the readers to understand how your EQE/WPE was measured exactly will help calibrate their thinking and appreciate your work more. For example, if EQE is measured on wafer, it is possible to get a lower EQE value compared to EQE measured within an integrating sphere.

7.       Line no. 145 - Reference 30 mentions: the indium content fluctuations resulting in fluctuations of bandgap resulting in different potential minima cause the FWHM broadening which is not explained by the authors in the current manuscript that well. "Band gap narrowing" is not a correct explanation, so please add a more descriptive phrase with the reference 30, to help reviewer understand the reason without going to the paper in ref. 30, which I had to do, but regular readers would not like to do that.

8.       Overall, few more citations (5-7 more) need to be added from across the world especially by leading teams in Europe, Japan, and USA who have performed pioneering work in scaling blue micro-LEDs in the last 2-3 years. 

Reviewer 2 Report

The paper presents a high resolution AM MicroLED display for potential application in near eye displays. The authors proposed a novel self-aligned mesa formation to achiveve the goal. The high resolution display performance was well investigated and characterized. The work could be considered for publication after addressing the following comments.

(1) In the introduction part, authors have cited the related publications concerning high resolution MicroLED display. It is better to comment on the scientific or technical issues which hindered the process toward high resolution MicroLED display.

(2) It is suggested that the authors could provide some discussions on the resolution requirements from the perspective of near eye applications.

Reviewer 3 Report

In this manuscript, the authors reported an active-matrix micro light-emitting diode (Micro-LED) display with 8 a resolution of 1920×1080 and high pixel density of 3200 pixels per inch. I suggest its acceptance after the following issues have been addressed.

1.     In introduction,the novelty of Micro-LED should be clarified more clearly. Some resent works such as Nanomaterials, 2022, 12(4): 627.; ACS applied materials & interfaces, 2018, 10(6): 5641-5648. have missed to be cited.

2.     The model number of lithographic aligner should be supplemented

3.     It is suggested to supplement the concrete etching steps of upside down Trapezoid mesa, since it is difficult to achieve the etching of inverted trapezoid mesa by conventional etching method.

4.     The normal operating current or current density of the device is much higher than the VF, 22A/cm2, indicating that no heat sink is needed. This reason is not sufficient.

5.     How to account for high series resistance, which increases the power consumption of the device?

6.     How to consider the current density of normal operation is generally not the highest efficiency of the current density problem? That is, the current density when working normally, and the efficiency has a more obvious decline.

Round 2

Reviewer 3 Report

The authors have responded my concerns in the revised manuscript, so I would like to recommend its acceptance at this stage.